# Air Pollution Detection Using a Novel Snap-Shot Hyperspectral Imaging Technique

**DOI:** 10.3390/s22166231

**Published:** 2022-08-19

**Authors:** Arvind Mukundan, Chia-Cheng Huang, Ting-Chun Men, Fen-Chi Lin, Hsiang-Chen Wang

**Affiliations:** 1Department of Mechanical Engineering, Advanced Institute of Manufacturing with High Tech Innovations (AIM-HI), Center for Innovative Research on Aging Society (CIRAS), National Chung Cheng University, 168, University Rd., Min Hsiung, Chiayi City 62102, Taiwan; 2Ophthalmology, Kaohsiung Armed Forces General Hospital, 2, Zhongzheng 1st. Rd., Lingya District, Kaohsiung City 80284, Taiwan

**Keywords:** hyperspectral imaging technology, air pollution, PM2.5, deep neural network, 3D convolutional neural network, auto encoding

## Abstract

Air pollution has emerged as a global problem in recent years. Particularly, particulate matter (PM2.5) with a diameter of less than 2.5 μm can move through the air and transfer dangerous compounds to the lungs through human breathing, thereby creating major health issues. This research proposes a large-scale, low-cost solution for detecting air pollution by combining hyperspectral imaging (HSI) technology and deep learning techniques. By modeling the visible-light HSI technology of the aerial camera, the image acquired by the drone camera is endowed with hyperspectral information. Two methods are used for the classification of the images. That is, 3D Convolutional Neural Network Auto Encoder and principal components analysis (PCA) are paired with VGG-16 (Visual Geometry Group) to find the optical properties of air pollution. The images are classified into good, moderate, and severe based on the concentration of PM2.5 particles in the images. The results suggest that the PCA + VGG-16 has the highest average classification accuracy of 85.93%.

## 1. Introduction

Many contemporary studies proved that air pollution caused many different diseases in human beings [1,2,3]. Air pollution has been a predominant cause of cardiovascular and respiratory diseases, and sometimes even the malfunctioning of the lungs [4,5,6]. In the past, most studies inferred that particulate matter 2.5 (PM2.5) has a direct effect on the mortality rate. The diameter of PM2.5 is less than 2.5 μm; therefore, it can penetrate through the lungs and completely enter the blood vessels, spreading through the human body [7,8,9]. Therefore, knowing the concentration of PM2.5 in the environment is a necessity to take basic precautions.

Most of the current state-of-the-art methods use spot testing methods to extract the air from the surroundings to check for the gases present in it. In previous studies, open-path Fourier-transform infrared (OP-FTIR) spectroscopy has been extensively used by various authors to detect suspended particles in the atmosphere [10,11,12,13,14]. Chang et al. [15] combined OP-FTIR with principal component analysis (PCA) to determine the source of the volatile organic compounds (VOCs) in an industrial complex. However, the level of the signal-to-noise ratio in most of the OP/FTIR systems is very low because of the poor collimation ability, thereby affecting the results [16]. Ebner et al. [17] proposed another method. They used a quantum cascade laser (QCL) open-path system, which overcame the disadvantages of OP-FTIR. Yin et al. [18] also utilized QCL to measure the parts per billion value (ppb) of SO_2_ in the mid-IR range. Zheng et al. [19] designed a tunable laser adsorption spectroscope using QCL to determine the amount of NO. However, the QC lasers are expensive and, therefore, not a popular option for detecting air pollution. In recent years, machine learning and deep learning models have been used to detect air pollution. One such study by Ma et al. used six different models based on the MLR, kNN, SVR, RT, RF and BPNN algorithms individually to find out the PM2.5 concentration. However, the real measurement data required to validate the correctness of the retrieval findings at a resolution of 30 m are missing from the equipment [20]. Another study by Zhang et al. used CNN to predict and forecast the hourly PM 2.5 concentration [21]. Although the model increased the prediction performance, the study did not consider the effect of regulatory policy for air control.

One method which could overcome the aforementioned disadvantages is hyperspectral imaging (HSI) engineering. Apart from the various applications of HSI, it has also been used to detect air pollution in recent years [22,23,24,25,26,27]. Chan et al. [28] set up multi-axis, differential optical absorption spectroscopy to verify the measurements of the oxygen measurement instruments of NO_2_ and HCHO over Nanjing. Jeon et al. [29] used hyperspectral sensors to estimate the concentrations of NO_2_ coming out of the air pollutants from the industry. However, most methods have used sensors and other instruments, which are expensive and require heavy instruments. Most hyperspectral applications have remained only at the laboratory level [30].

Therefore, this study combines HSI technology and deep learning techniques to propose a large-scale, low-cost air pollution detection method. Three models have been developed: 3D Convolutional Neural Network Auto Encoder, principal components analysis (PCA), and the Red, Green, Blue (RGB) images, which are, respectively, proposed and combined with VGG-16 (Visual Geometry Group) to classify the images into three categories: good, moderate, and severe, based on the concentration of PM2.5. Finally, the accuracy of each of the methods was compared to determine the optimal model.

## 2. Materials and Methods

### 2.1. Dataset

As no suitable PM2.5 image dataset exists, this study created a dataset using an aerial camera DJI MAVIC MINI from 9:00 a.m. to 5:00 p.m. every day at an interval of an hour. The images were taken at the height of 100 m on the Innovation Building of National Chung Cheng University at an angle of 90°. The images have been scaled to make the input image match the input size of the first layer of VGG-16 to 224 × 224 in pixels. The data from PM2.5 of the Environmental Protection Agency’s Pakzi Monitoring Station were used as a reference to classify the images into three categories: good, moderate, and severe, according to the severity of air pollution. The total number of images in the dataset was 3340 images. In this study, the data were divided into training, validation, and test sets according to the ratio of 6:2:2, respectively.

### 2.2. HSI Algorithm

HSI has been recently used in many industries including agriculture [31], astronomy [32], military [33], biosensors [34,35,36], remote sensing [37], dental imaging [38], environment monitoring [39], satellite photography [40], cancer detection [27,41], forestry monitoring [42], food security [43], natural resources surveying [44], vegetation observation [45], and geological mapping [46]. The visible-light hyperspectral imaging (VIS-HSI) developed in this study uses an aerial camera (DJI MAVIC MINI) image to convert the image into a hyperspectral image in the visible wavelength range of 380–780 nm and up to a spectral resolution of 1 nm. The relationship matrix between the camera and the spectrometer has to be found to construct the VIS-HSI algorithm, as shown in Figure 1.

The camera (DJI MAVIC MINI) and the spectrometer (Ocean Optics, QE65000) must be given multiple common targets as analysis benchmarks, which will greatly improve the accuracy. Thus, the standard 24-color checker (x-rite classic, 24-color checker) was selected as the target because it contains the most important and all the common colors. As the camera may be affected by inaccurate white balance, the standard 24-color card must be passed through the camera and the spectrometer to obtain 24-color patch images (sRGB, 8bit) and 24-color images, respectively. The 24-color patch image and 24-color patch reflection spectrum data are converted to CIE 1931 XYZ color space (refer to Appendix A for individual conversion formulas). In the camera part, the image (JPEG, 8bit) is stored according to the sRGB color-space specification. Before converting the image from the sRGB color gamut space to the *XYZ* color gamut space, the respective R, G, and B values (0–255) must be converted to a smaller scale range (0–1). Using the gamma function, the sRGB value is converted into a linear RGB value, and finally, through the conversion matrix, the linear RGB value is converted into the standard in the *XYZ* color gamut space. In the spectrometer part, to convert the reflection spectrum data (380–780 nm, 1 nm) to the *XYZ* color gamut space, the *XYZ* color matching functions and the light-source spectrum *S(**λ)* are required. Brightness is calculated from the *Y* value of the *XYZ* color gamut space as both values are proportional. The brightness value is normalized between 0 and 100 to obtain the luminance ratio k, and finally, the reflection spectrum data are converted to the *XYZ* value (*XYZ_Spectrum_*).

The variable matrix *V* is obtained by analyzing the factors that may cause errors in the camera, such as the nonlinear response of the camera, the dark current of the camera, inaccurate color separation of the color filter, and color shift. Regression analysis is performed on *V* to obtain the correction coefficient matrix *C* used to correct the errors from the camera, as shown in Equation (1). The average root-mean-square error (RMSE) of the data of *XYZ_Correct_* and *XYZ_Spectrum_* was found to be only 0.5355. Once the calibration process is completed, the obtained *XYZ_Correct_* and the reflection spectrum data of the 24 color patches (*R_Spectrum_*) measured by the spectrometer are compared. The objective is to obtain the conversion matrix *M* by finding the important principal components of RSpectrum through PCA and multiple regression analysis, as shown in Equation (2). In the multivariate regression analysis of *XYZ_Correct_* and Score, the variable *VColor* is selected because it has listed all possible combinations of X, Y, and Z. The transformation matrix *M* is obtained through Equation (3), and *XYZ_Correct_* is used to calculate the analog spectrum (*S_Spectrum_*) through Equation (4).
(1)$$\left[C\right]=\left[XYZ_{Spectrum}\right]\times \mathrm{pinv}\left(\left[V\right]\right),$$
(2)$$\left[XYZ_{Correct}\right]=\left[C\right]\times \left[V\right],$$
(3)$$\left[M\right]=\left[Score\right]\times pinv\left(\left[V_{Color}\right]\right),$$
(4)$${\left[SSpectrum\right]}_{380\sim 780\;nm}=\left[EV\right]\left[M\right]\left[V_{Color}\right].$$

Finally, the obtained analog spectrum of 24 color blocks (*S_Spectrum_*) is compared with the reflection spectrum of 24 color blocks (*R_Spectrum_*). The RMSE of each color block is calculated, and the average error is 0.0532. The difference between SSpectrum and RSpectrum can also be represented by the color difference. The VIS-HSI algorithm can be established through the above process and can simulate the reflection spectrum of the RGB values captured by the camera.

### 2.3. Three-Dimensional-Convolution Auto Encoder

The hyperspectral image converted from the original RGB image can be represented as *X*, where *X = h × w × b*, *b* is the number of input channels, and *h × w* is the size of the input image. The first layer L1 receives the input image *X*, whereas the last layer LN is the output layer. The intermediate layers include seven convolutional layers, three pooling layers, and one fully connected layer. In the proposed model, 3D-CNN is used for compression because, unlike 2D-CNN, using the spatial convolution kernel can learn both spatial and spectral features. After training, the compressed data will be used as the input layer of VGG-16 to continue training. The problem of data sparsity in high dimensions is solved using feature-dimension reduction or band selection of frequency bands. This study uses PCA. The dimensionality reduction method of the 3D convolutional autoencoder (3D-CAE) preprocesses the data to reduce the number of frequency bands and finally sends the dimensionally reduced images to VGG-16 for training. The input data can be expressed as (*n*, *X*), where *n* is the number of samples. The dimensions of the spectral features are reduced, and the data are organized in the form of (*n × h × w × b*) to perform PCA. The whole methodology developed in this study is shown in Figure 2.

## 3. Results and Discussion

This study uses PCA in machine learning and the 3D-CAE model in deep learning for feature dimensionality reduction. The results of the study show that only three principal components are used to represent 99.87% of the data, retaining most of the information from the original data. The 3D-CAE model proposed in this study can jointly learn 2D spatial features and 1D spectral features from HSI data, and the compressed hyperspectral images. Such images can greatly reduce the data size and avoid the problem of insufficient memory space. The loss function and the accuracy of the results show that the loss has gradually converged when the fourth epoch is trained. The epoch refers to the process in which complete data pass through the neural network once and return once. As the epoch increases, the number of weight updates in the neural network will also increase (see Appendix A for the training accuracy of all three models). The classification results will classify the images into three categories: good, moderate, and severe, according to the severity of air pollution.

A total of four indicators were used to judge the performance of the three models. Precision, also known as the positive predicted value, represents the number of truly positive cases out of all the predicted positive cases as shown in Equation (5). Recall rate, also known as sensitivity, represents the amount of predicted positive cases out of all the positive cases as shown in Equation (6). The F1 score is the harmonic mean of precision and recall. This value is high on an imbalanced dataset as shown in Equation (7), while accuracy of the predicted model can be calculated by dividing the total correct prediction and the total dataset as shown in Equation (8).
(5)$$\mathrm{Precision}=\frac{\mathrm{TP}}{\mathrm{TP}+\mathrm{FP}}$$
(6)$$\mathrm{Recall}=\frac{\mathrm{TP}}{\mathrm{TP}+\mathrm{FN}}$$
(7)$$F1\mathrm{-score}=\frac{2\;\times \mathrm{Recall}\;\times \mathrm{Precision}}{\mathrm{Recall}\;+\mathrm{Precision}}$$
(8)$$\mathrm{Accuracy}=\frac{\mathrm{TP}\;+\mathrm{TN}}{\mathrm{TP}+\mathrm{FP}+\mathrm{FN}+\mathrm{TN}\;}$$

Figure 3 represents the confusion matrix of the three methods. The confusion matrix shows 210 good, 168 moderate, and 149 severe fully classified correctly in the RGB image. In the PCA + VGG16 method, 240 good, 155 moderate, and 179 severe were correctly classified, whereas 246 good, 22 moderate, and 1 severe were classified correctly in 3D-CAE images. Table 1 shows the accuracy of every method. The F1-score of moderate is lower because this category was more difficult to judge. The results show that the PCA + VGG-16 model has the best classification accuracy, followed by RGB + VGG-16, and finally 3D-CAE + VGG-16, as shown in Table 2. The 3D-CAE + VGG-16 has poor results because the compressed features do not necessarily have the optical properties of PM2.5.

The main practical applicability of this study lies in its future scope, where a smart phone with a camera can be endowed with hyperspectral imaging technology. A mobile application can be developed, which could actually let the user know about the concentration of PM2.5 particulates in a specific image. In this method, the detection of air pollution in the atmosphere can be mobilized, thereby reducing the number of apparatuses required and the complexity of measurement. This method can also be extended to measure the PM10 concentration in the environment.

## 4. Conclusions

In this study, new VIS-HSI technology has been combined with artificial intelligence to estimate PM2.5 concentration in the images captured from the drone. Three algorithms have been developed. Through the dimensionality reduction methods of PCA, 3D-CAE, and VGG-16 neural network, the images are divided into three categories based on the PM2.5 concentration. In terms of experimental accuracy, PCA dimensionality reduction is the most effective, followed by the RGB method, and finally the 3D-CAE method. The accuracy of the methods can be improved by increasing the dataset that contains PM2.5 concentrations or fine-tuning the models. By integrating band selection with the CNN model, the accuracy of the prediction models can be improved significantly. Other weather features, such as wind speed, humidity, and temperature, can be considered to predict air pollution. Apart from PM2.5 concentrations, the prediction models can also be designed to predict PM10 concentrations. In the future, the same technology can be integrated with the cameras of the smartphone to detect real-time air pollution on the spot.

## Figures and Tables

**Figure 1 sensors-22-06231-f001:**
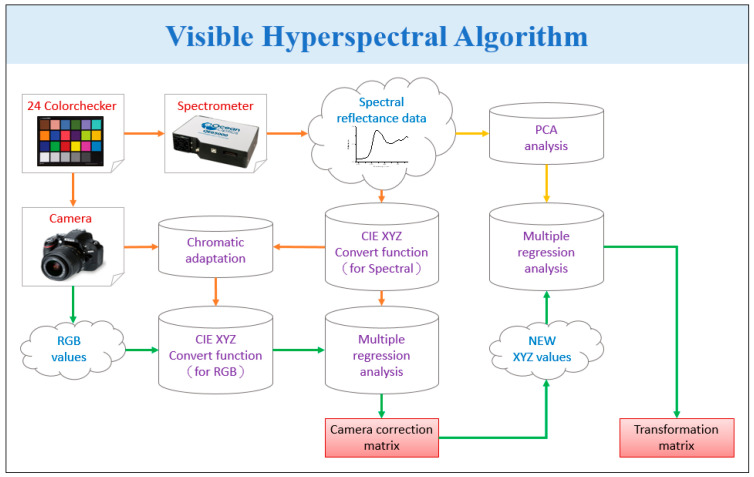
Visible Hyperspectral Imaging Algorithm.

**Figure 2 sensors-22-06231-f002:**
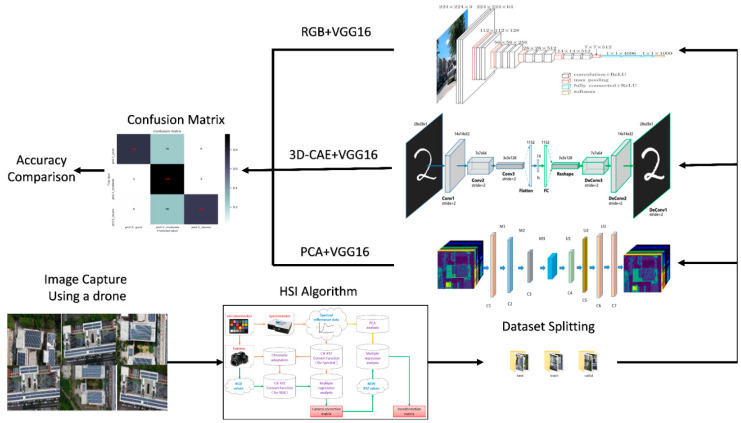
The methodology used in this study.

**Figure 3 sensors-22-06231-f003:**
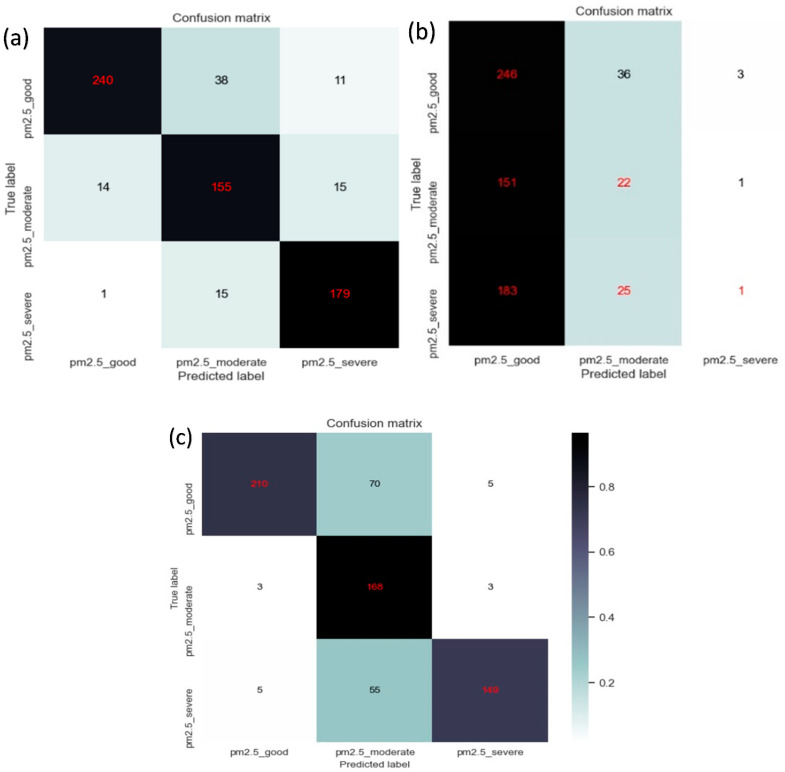
Confusion matrix of the methods developed. Panels (**a**–**c**) represent the PCA method, the 3D-CAE method, and the RGB method, respectively.

**Table 1 sensors-22-06231-t001:** Shows the Precision, Recall and F1 score of all the three models.

**RGB**	**Precision**	**Recall**	**F1 Score**
Good	96.33%	73.68%	83.50%
Moderate	57.34%	96.55%	71.95%
Severe	94.90%	71.29%	81.42%
**PCA**	**Precision**	**Recall**	**F1 Score**
Good	94.12%	83.04%	88.24%
Moderate	74.52%	84.24%	79.08%
Severe	87.32%	91.79%	89.50%
**3D-CAE**	**Precision**	**Recall**	**F1 Score**
Good	42.41%	86.32%	56.88%
Moderate	26.54%	12.64%	17.12%
Severe	20.00%	0.48%	0.93%

**Table 2 sensors-22-06231-t002:** Classification accuracy of the three methods proposed in the study.

Method	Classification Accuracy (%)
PCA	85.93
RGB	78.89
3D-CAE	40.27

## Data Availability

Not applicable.

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
