# Peer review of "Air Pollution Detection Using a Novel Snap-Shot Hyperspectral Imaging Technique"

_sensors, 2022, doi:10.3390/s22166231_

Round 1

Reviewer 1 Report

In this manuscript, the authors propose a large-scale, low-cost solution for detecting air pollution by combining hyperspectral imaging technology and deep learning. By modeling the visible-light hyperspectral imaging technology of the aerial camera, the image acquired by the drone camera is endowed with hyperspectral information. Two methods (3D Convolutional Neural Network Auto Encoder and principal components analysis) are used for the classification of the images, and the results are also evaluated.

Some suggestions to improve this manuscript:

(1) Section 2:  I suggest adding 1 or 2 figures and corresponding expressions to display the selected spectral reflectance data (380-780 nm with the 1nm spectral resolution) more intuitively in this manuscript. For the details, the authors can refer to the format of Fig. 6 (a-b) in
*https://www.google.com/url?q=https://www.sciencedirect.com/science/article/abs/pii/S0022407320302004?via%253Dihub*&source=gmail-imap&ust=1659680524000000&usg=AOvVaw0zscYw4V_azsqlzfHsVVqL.

(2) Section 3, “The results of the study show that only three principal components are used to represent 99.87% of the data”: For the PCA, I also suggest adding 1 figure to display the real PCA results. For example, the each and cumulative variance contribution (%) as a function of principal component number.

(3) Line 176-177: The Fig. No. “figure 2” should be corrected to “figure 3”. Besides, this figure is not clear, and needs to be further improved.

Reviewer 2 Report

This paper deals with a new technique (drone´s image) to investigate the air pollution (PM2.5) using math/statistical techniques. I did not understand properly what was the truth values used to validate the trained values from ML. This point should be clear for the final version, on my opinion. The attached version has others comments/suggestions.

Reviewer 3 Report

This paper proposes a novel snap-shot hyperspectral imaging technique for air pollution detection. The experimental idea is relatively novel, but there are still some problems.

1) In the introduction section, a summary of the integrated image and deep learning methods for PM2.5 detection available in previous studies is inadequate.

https://doi.org/10.3390/rs14030599

https://doi.org/10.1016/j.cageo.2021.104869

The above pieces of literature are the latest published which is related to related conception. Would you mind comparing them in the introduction section, and elaborating your technical advantages?

2) The results and discussion section is slightly weak, listing only the accuracy of the three models without stating the applicability and practical applications of the three models.

3) A variety of methods for fine-grained PM2.5 inversion already exist for current research. The method in this paper only classifies PM2.5 into three categories, what is its practical application?

Reviewer 4 Report

This manuscript presents the results of an air quality classification system based on the optical properties of air based on the evaluation of images obtained with cameras installed on drones.

The novelty in the work is in the combination of hyperspectral imaging and deep learning; the idea is to have a "low-cost" air pollution detection method. It was not clear how this would be "low-cost."

The introduction is very good with relevant literature cited and put into perspective. The acronym QCL on page 2, line 44, should be defined [quantum cascade laser] before it is used.

The materials and methods section clearly presents the detals of the Dataset, the HSI Algorithm and the Auto Encoder.

The results and discussion section presents the results of the work in enough details. Equations (1) to (4), lines 171 to 174, should be re-numbered as Equations from (5) to (8) as a continuation of the equations from page 4 (lines 122 to 125). Also the variables TP, FP, FN, and TN should be defined (lines 171 to 174).

On page 6, line 163, "The F1-score of moderate is lower because this category was more difficult to judge." Why is this the case? Also from Table 1 this is only the case for the RGB and PCA methods, but not for the 3D-CAE method.

On page 6, line184, "The results show that the PCA+VGG-16 model has the best classification accuracy,...." It should be clear that this refers to Table 2 on page 7.

On page 6, line 186, "3D-CAE+VGG-16 has poor results because the compressed features do not necessarily have the optical properties of PM2.5." Could this statement be backed up with facts/data/references?

Very good conclusions.

Round 2

Reviewer 1 Report

Since the authors have modified the manuscript and added the necessary supplementary by my suggestions, I don't have other comments any more.

Reviewer 3 Report

All the suggestions have been well modified.